# The Removal of Subterranean Stormwater Drain Sumps as Mosquito Breeding Sites in Darwin, Australia

**DOI:** 10.3390/tropicalmed5010009

**Published:** 2020-01-10

**Authors:** Allan Warchot, Peter Whelan, John Brown, Tony Vincent, Jane Carter, Nina Kurucz

**Affiliations:** 1Medical Entomology, Centre for Disease Control, Top End Health Service, Northern Territory Government, Darwin, Northern Territory 0810, Australia; jane.carter@nt.gov.au (J.C.); nina.kurucz@nt.gov.au (N.K.); 2Biting Insect Technical Extension Services, Nightcliff, Darwin, Northern Territory 0810, Australia; peteriwhelan@bigpond.com; 3Civil Infrastructure, City of Darwin, Darwin, Northern Territory 0810, Australia; j.brown@darwin.nt.gov.au (J.B.); t.vincent@darwin.nt.gov.au (T.V.)

**Keywords:** exotic mosquitoes, sump, subterranean storm water drain, maintenance breeding site, rectification

## Abstract

The Northern Territory Top End Health Service, Medical Entomology Section and the City of Darwin council carry out a joint Mosquito Engineering Program targeting the rectification of mosquito breeding sites in the City of Darwin, Northern Territory, Australia. In 2005, an investigation into potential subterranean stormwater breeding sites in the City of Darwin commenced, specifically targeting roadside stormwater side entry pits. There were 79 side entry pits randomly investigated for mosquito breeding in the Darwin suburbs of Nightcliff and Rapid Creek, with 69.6% of the pits containing water holding sumps, and 45.6% of those water holding sumps breeding endemic mosquitoes. *Culex quinquefasciatus* was the most common mosquito collected, accounting for 73% of all mosquito identifications, with the potential vector mosquito *Aedes notoscriptus* also recovered from a small number of sumps. The sumps were also considered potential dry season maintenance breeding sites for important exotic *Aedes* mosquitoes such as *Aedes aegypti* and *Aedes albopictus*, which are potential vectors of dengue, chickungunya and Zika virus. Overall, 1229 side entry pits were inspected in ten Darwin suburbs from 2005 to 2008, with 180 water holding sumps identified and rectified by concrete filling.

## 1. Introduction

Darwin is the capital city of the Northern Territory (NT), Australia, and is situated in the wet-dry tropics with an average annual rainfall of 1722.6 mm [1]. Almost all rainfall in Darwin is recorded in the wet season between November to April, with very little rainfall occurring during the dry season from May to October [1]. Due to the constant high temperatures and high seasonal rainfall, and the proximity of Darwin to low lying coastal floodplains, mosquitoes are an ongoing year-round problem, both from a pest perspective, and as potential vectors of human disease. The most important mosquito borne diseases in the NT are those caused by Murray Valley encephalitis virus (MVEV), Kunjin virus (KUNV), Ross River virus (RRV), and Barmah Forest virus (BFV), with RRV being by far the most common [2].

To reduce the burden of mosquito borne disease, the NT has active mosquito monitoring and control programs that service the major residential centers [3], with the highest control efforts concentrated around Darwin, due to the denser population and location adjacent to localised and extensive coastal wetlands, and due to the direct sea and air transport links with Asia [4]. Darwin also has a unique combined Mosquito Engineering Program between the NT Government, Top End Health Service Medical Entomology Section, and the City of Darwin (CoD) local council [3,4], which was created in 1982 to rectify the widespread mosquito problems caused by stormwater discharge along the coastal fringes of urban Darwin [5].

The NT is currently free of the dengue vector mosquito *Aedes aegypti*, however this species was present in the NT at least until 1956 [3]. Historical records documented dengue infection in Australian aborigines [6] and outbreaks in troops during WWII [7]. It is thought that *Ae. aegypti* disappeared due to a reduction of artificial breeding receptacles following the movement to reticulated water and diesel trains [2,8]. However, in the past 13 years, *Ae. aegypti* incursions occurred in Tennant Creek on two separate occasions in 2004 and 2011, and once on Groote Eylandt during 2006, with all three incursions eliminated via intensive 2–3 year programs [9,10,11]. Imported *Aedes aegypti* and the Asian tiger mosquito *Aedes albopictus* are also frequently found in Department of Agriculture and Water Resources surveillance traps at Darwin international air and seaports [12]. This highlights the potential for exotic mosquito establishment in the NT, and the subsequent exotic disease risk of not only dengue, but other diseases known to be carried by these mosquitoes such as chickungunya [13] and Zika viruses [14]. 

Drain sumps are considered potential larval habitats for *Ae. aegypti* [15,16] and *Ae. albopictus* [17,18,19], indicating the potential for drain sumps in Darwin to act as subterranean larval habitats for these exotic *Aedes* mosquitoes should an incursion occur. It was not commonly thought that drain sumps were present in Darwin, due to the general requirement for free draining stormwater pipe systems in the city of Darwin. However, the identification of a water holding side entry pit sump in the suburb of Rapid Creek in 2005 suggested there were areas of Darwin that contained stormwater sumps. The evidence of stormwater sumps in one Darwin suburb, and the potential for stormwater sumps to act as larval habitats for endemic and exotic mosquitoes, resulted in the combined Mosquito Engineering Program targeting the inspection and remediation of stormwater sumps the City of Darwin from 2005 to 2007. 

## 2. Materials and Methods 

An initial inspection for the presence of roadside entry pit sumps for mosquito breeding was carried out during the dry season of 2005 in Nightcliff and Rapid Creek. These two residential suburbs are located in the Darwin northern suburbs, and were predominantly built in the 1950s and 1960s on relatively large lots (800–1000 m^2^), and therefore the suburbs are characterised by established gardens with large shading trees. Due to the long Darwin dry season and deciduous nature of many native trees, large amounts of leaf litter can enter the underground stormwater system, adding a potential source of nutrients. Dry season water can enter the stormwater drains via the overwatering of road verges, washing of cars, boats and domestic bins on driveways, irrigation of school grounds and sporting fields, and cleaning of swimming pools, with the low flow water and vegetation debris potentially producing suitable conditions for mosquito breeding in subterranean drains (Figure 1a). 

Side entry pits are stormwater inlet pits located along the side of the road, and are the most common type of road stormwater inlet pit in the City of Darwin. Lids of the side entry pits are made of pre-cast concrete, usually fitted with two small lift holes, and therefore able to be manually lifted by council staff (Figure 1b). The general dimensions of the side entry pits tends to vary, however most pits surveyed were generally less than 1m long by about 40–60 cm wide.

All side entry pit locations were identified and plotted on field maps using CoD stormwater pits and the NT Government road and lot boundary ArcGIS^®^ shapefile data. In 2005, Medical Entomology or City of Darwin staff randomly surveyed 79 side entry pits in Nightcliff and Rapid Creek by lifting the pit lids, and dipping with 350 ml volume ABC dipper with telescoping handle (Pacific Biologics^®^, Kippa-Ring, Australia). Dips were taken in the corners of any water holding sump, and amongst floating debris and rubbish. A maximum of 5 dips were taken, with a wait of one minute between dips to allow some larvae that had dived to resurface. The aim of the dipping was to identify species diversity, rather than estimate total numbers of larvae. All collected mosquito larvae were stored in small sample vials and preserved in 70% alcohol for later identification under a light microscope. 

Once the 79 pits had been sampled, the program focus shifted to the location and rectification of stormwater sumps in the remaining areas of Nightcliff and Rapid Creek. The rectification program then expanded into other Darwin suburbs during 2006 to 2008. Lids of inspected pits that contained water holding sumps were marked with a yellow dot to allow easy locating of the problem pits by the rectification team. Rectified pits were marked with a pink dot.

## 3. Results

### 3.1. Mosquito Sampling

The results of the 79 side entry pit sumps investigated in the suburbs of Nightcliff and Rapid Creek, are shown in Table 1. The location of all pits surveyed and found with mosquito larvae is shown in Figure 2. Two thirds of the pits inspected in Nightcliff were positive for mosquito larvae (67%), while 38% of the pits in Rapid Creek were positive for mosquito larvae (Table 1). Overall, 90% of side entry pit in Nightcliff and 62% of side entry pit in Rapid Creek were discovered with water holding sumps during the investigation (Table 1). 

There were a total of 45 mosquito species identifications from the 36 sumps found positive for mosquito larvae (Table 2). The reason for more identifications than actual positive sumps was due to the co-habitation of species in some of the sumps. The most common larval mosquito collected was *Culex quinquefasciatus,* which represented 73% of all mosquito identifications in the sumps. A total of 33 of the 36 sumps positive for mosquito breeding contained this species. Other mosquito species were uncommon, and included the predacious mosquito larvae *Culex halifaxii*, which was identified in three of the positive sumps, as well as *Culex pullus*, which was identified in one of the positive sumps. The only *Aedes* species recovered during the surveys was *Aedes notoscriptus,* which was identified in two sumps located in Nightcliff. 

### 3.2. Inspections for Water Holding Sumps

Between 2005 and 2008, a total of 1229 side entry pits were inspected during the dry season for the presence of a water holding sump across 10 suburbs in Darwin (Table 3). All pits with actual or potential water holding sumps were earmarked for rectification. Only three Darwin suburbs contained pits with sumps; the northern suburbs of Nightcliff, Rapid Creek and Millner. The number of pits with sumps varied across the three suburbs, with the greatest number of sumps located in Rapid Creek, with Rapid Creek also having a high percentage of pits that contained a sump (Table 3). 

### 3.3. Sump Rectification

Once stormwater pits were identified as capable of having water holding sumps, they were vacuumed of debris and rubbish via a jet/vacuum truck (Figure 3a,b), measured to determine the sump depth, and hand filled and levelled with concrete to be free draining, thus permanently removing their ability to be larval mosquito habitats. All rectification works were carried out during the dry season, with supervisors checking to ensure the pits were appropriately levelled to be completely free draining. Subsequently, no follow up larval mosquito surveys were carried out due to the assumption that all pits had been rectified of potential water holding.

## 4. Discussion

The side entry pit rectification program carried out between 2005 and 2007 was successful in removing a large number of dry season larval habitats in three suburbs of Darwin. The 2005 mosquito survey showed that the majority of side entry pit sumps contained water during the dry season, with almost half of the water holding sumps positive for mosquito larvae. The sumps were not considered potential wet season larval habitats, due to the frequent wet season flushing events.

*Culex quinquefasciatus* was by far the most common mosquito species collected in the sumps, due to high nutrient water caused by anthropogenic water runoff and organic matter and rubbish. Past studies have revealed that *Cx. quinquefasciatus* is generally considered an unlikely vector of human disease in Australia [20]. However, this mosquito does have significance as one of the principal vectors of West Nile virus in the US [21], and a vector of filariasis in Asia [22]. Vector competence trials on Australian *Cx. quinquefasciatus* populations revealed they are capable of transmitting an American strain of West Nile virus [23], therefore this species does have the potential to be a vector mosquito in Darwin. Overall, regardless of the vector importance of *Cx. quinquefasciatus*, the side entry pit rectification program has at least removed one type of dry season cryptic larval habitat from those three Darwin suburbs.

Common dry season *Cx. quinquefasciatus* larval habitats in Darwin are generally considered to be open unlined drains with polluted water, blocked stormwater drain pipes, and backyard containers such as disused swimming pools, bird baths, plant strike buckets and pot plant drip trays. The side entry pit sumps were identified as an additional larval habitat during the 2005 survey. Due to the operational nature of the Mosquito Engineering Program, there was no evaluation of the overall impact that these sumps may have had on adjacent residents with regards to creating a mosquito problem. In the Northern Territory, current public health legislation determines that the landholder is responsible for preventing or controlling mosquito breeding. As soon as the stormwater sumps were identified as larval mosquito habitat, resources were committed to the expedient removal of those sumps, to remove any requirement to carry out monitoring and control. It was generally viewed that larval mosquito control of the sumps would be burdensome to manage. The overall benefit of the program was primarily the removal of one type of cryptic dry season larval habitat for *Cx. quinquefasciatus*, with some reduction in adult numbers expected to have occurred, at least in Rapid Creek where the largest numbers of sumps were present. 

Whilst only two sumps were positive for *Ae. notoscriptus,* this mosquito species is a potential vector of Ross River virus in the NT [24,25], and therefore the removal of man-made larval habitats is considered important. Common dry season larval habitats in Darwin are generally considered to be pot plant drip trays, plant strike buckets, dog bowls and bird baths. The side entry pit sumps were identified as an additional larval habitat during the 2005 survey. However, the limited prevalence of *Ae. notoscriptus* during the survey suggest the sumps were not important dry season larval habitats, possibly due to generally putrid water, for which *Ae. notoscriptus* is not usually associated with in Darwin. In the absence of further larval sampling data from the sumps before rectification occurred, and data from adjacent residential yards, the removal of one type of cryptic larval habitat is considered the main benefit of the rectification program with regards to *Ae. notoscriptus*.

With regards to the potential for the side entry pit sumps to be important dengue mosquito larval habitats if an incursion was to occur, it is possible that the sumps would have served as establishment and maintenance larval habitats in the prolonged Darwin dry season, allowing the rapid colonisation of adjacent urban backyards upon the commencement of the wet season. A laboratory study in North Queensland found a long period of viability for *Ae. aegypti* eggs, with 2–15% of eggs viable after one year, and viability remaining above 88% under all investigated climate conditions through 56 days [26]. However, in the field in Townsville, North Queensland, all eggs were dead in subterranean locations after two months due to predation and fungal infection [16]. Therefore, the authors of the Townsville study concluded that it was the presence of stable larval populations in subterranean refuges that set up the colonisation of other breeding sites at the commencement of the wet season. Townsville has a similar wet-dry climate to Darwin, and therefore the conclusions from this study are likely to be applicable to Darwin, if the exotic *Ae. aegypti* was introduced. As drain sumps are also considered larval habitats for *Ae. albopictus* in other parts of the world, there is the potential that the side entry pit sumps in Darwin may have served as maintenance larval habitats for this species during the Darwin dry season.

The reason for the presence of the sumps in Rapid Creek, Nightcliff and Millner appears to have been due to no concrete floor being provided during initial construction, with the sumps created by long term scouring flows, as all side entry pit sumps that required rectification were noticed by council staff to have earthen floors. The depths of the sumps also varied from 5 to 400 mm deep, suggesting they were caused by erosion rather than constructed to a standard design. However, if the earthern sumps were provided for a reason, the design appeared to not be widely adopted and later abandoned. The side entry pits that had sumps in Rapid Creek and Millner were noted to have been constructed during 1960–1969, while most of the side entry pits in Nightcliff were constructed during 1955–1969. Two suburbs where side entry pits were built from 1964–1969 in the same catchment as Millner and Rapid Creek were Jingili and Moil, with no sumps located during the program in those suburbs. Later built roads in the Darwin suburbs of Coconut Grove, where side entry pits were built mostly from 1970–1974, and Marrara, where side entry pits were built during 1980–84, also did not contain side entry pits with sumps. This was also observed in Fannie Bay, Parap and Ludmilla, where some roads were built after 1969, but mostly from 1955–1964, did not contain side entry pit sumps. Overall, this suggests that the earthern sumps identified in the three Darwin suburbs were not a standard design for the city of Darwin. 

It is understood that in southern Australia, stormwater sumps serve a function to trap sediment. In the City of Darwin, to manage sediment runoff associated with high intensity rainfall events, large sediment ponds are provided during the construction phase at the drain outlet pipes, and the local drain authorities carry out routine dry season maintenance of drain outlet pipes in established suburbs, to ensure the pipe network remains free draining. This includes flushing the lower sections of pipes when required. Rubbish is collected via in-line net type gross pollutant traps situated at or near the outlet pipe. This ensures the underground stormwater system can be designed to be completely free draining. The potential maintenance difficulties with regards to subterranean devices most likely also influences the desire for a free draining stormwater pipe system. The positive flow on effect for mosquito authorities is preventing the creation of one type of cryptic larval mosquito habitat, which could be suitable for pest and potential disease carrying mosquito species that prefer small container type habitats.

The pro-active program to remove side entry pit sumps as potential larval mosquito habitat for exotic mosquitoes and endemic mosquitoes is testament to the commitment of the NT Government and City of Darwin council to remove anthropogenic created larval mosquito habitat around Darwin, as part of the combined Mosquito Engineering Program. The authors recommend that tropical cities with a similar high and intense rainfall climate also follow the City of Darwin council free draining stormwater sump design, and initiate rectification programs to rid their cities of potential mosquito breeding stormwater sumps.

## Figures and Tables

**Figure 1 tropicalmed-05-00009-f001:**
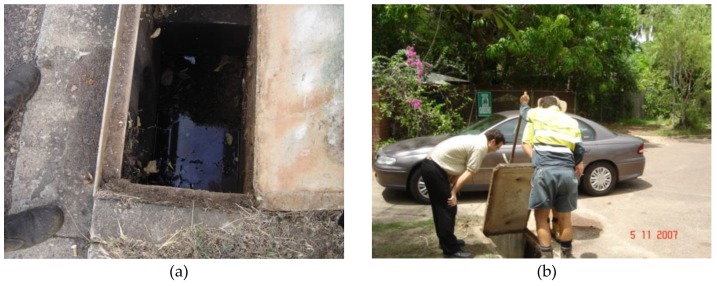
(**a**) Drain sump with dry season stagnant water in Rapid Creek; (**b**) inspecting drain pits.

**Figure 2 tropicalmed-05-00009-f002:**
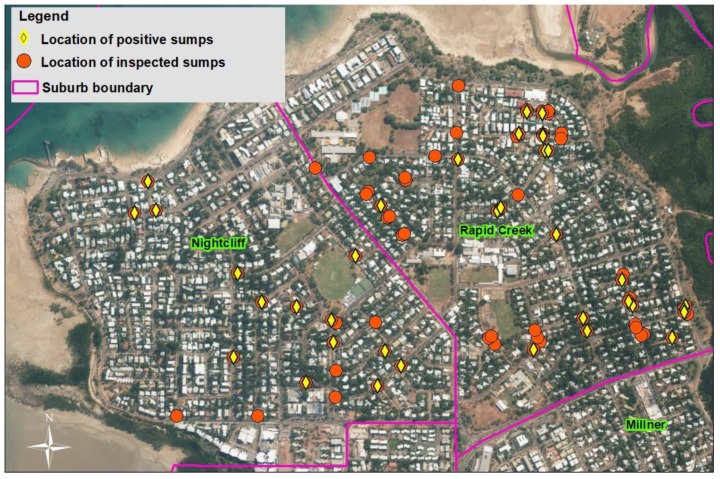
Location of surveyed side entry pits and sumps positive for mosquito larvae.

**Figure 3 tropicalmed-05-00009-f003:**
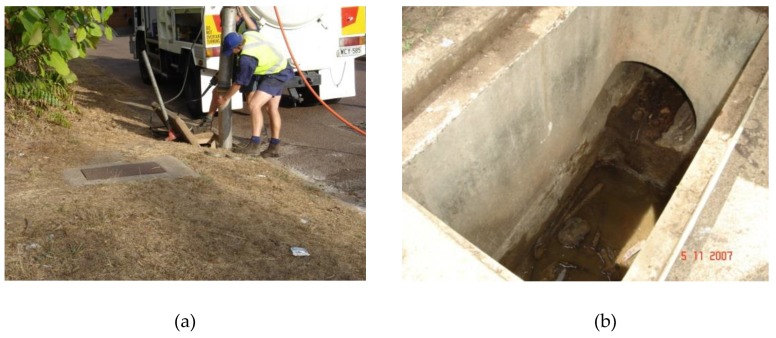
(**a**) Sump being cleaned by jet/vacuum; (**b**) cleaned sump ready for concrete filling.

**Table 1 tropicalmed-05-00009-t001:** Side entry pit sumps sampled during the 2005 dry season for the presence of water and mosquito larvae in Nightcliff and Rapid Creek, Darwin.

**Larvae Present**	**Nightcliff**	**Rapid Creek**	**Total**
Yes	14	22	36
No	7	36	43
Total	21	58	79
% breeding	66.7	37.9	45.6
**Water Present**	**Nightcliff**	**Rapid Creek**	**Total**
Yes	19	36	55
No	2	22	24
Total	21	58	79
% with water	90.5	62.1	69.6

**Table 2 tropicalmed-05-00009-t002:** Larval mosquito species identified in side entry pit sumps (n = 36).

Mosquito Species	Nightcliff	Rapid Creek	Total	%
*Aedes notoscriptus*	2	0	2	4.4
*Culicine pupae*	2	3	5	11.1
*Culex halifaxii*	1	2	3	6.7
*Culex pullus*	0	1	1	2.2
*Culex quinquefasciatus*	13	20	33	73.3
No sample	1	0	1	2.2
Total	19	26	45	100.0

**Table 3 tropicalmed-05-00009-t003:** Total number of side entry pits inspected by suburb, and number of pits with or without sumps by suburb. All sumps were rectified by concrete filling.

Suburb	Number with Sumps	Number without Sumps	Total	%
Coconut Grove	0	38	38	0.0
Fannie Bay	0	67	67	0.0
Jingili	0	227	227	0.0
Ludmilla	0	46	46	0.0
Marrara	0	22	22	0.0
Millner	23	120	143	16.1
Moil	0	297	297	0.0
Nightcliff	40	140	180	22.2
Parap	0	65	65	0.0
Rapid Creek	117	27	144	81.3
Total	180	1049	1229	14.6

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
