# Peer review of "The Removal of Subterranean Stormwater Drain Sumps as Mosquito Breeding Sites in Darwin, Australia"

_tropicalmed, 2020, doi:10.3390/tropicalmed5010009_

Round 1

Reviewer 1 Report

This paper is well written as it summarizes succinctly an activity to find and eliminate mosquito sources, here sumps associated with stormwater catchment systems.  The rationale for the work is clear and the research problem is clearly stated.  It is fair to say that this report is of an operational nature, that is, it exposes the diligent work of a control agency at a community level in detail.

There are two issues or concerns that the authors need to address.  Neither is "fatal" to the manuscript in terms of its acceptability, but information about them should be included in the manuscript to clarify the nature of the study.

The first is that this manuscript is clearly about an intended intervention, but there is no sampling study that compares the effect of the intervention before and after it was done.  The authors should comment about it, and make statements about effects of the intervention.  Otherwise, this study is really an operational activity with no evaluated outcome for the reader to be able to draw a conclusion about its effects.

The second is the nature and purpose of the sumps.  The authors relate in the Discussion that the origin of the sumps is not know and could have been generated by water flow over time (see lines 193-200).  I disagree with this section.  Sumps are typically purposefully built into these systems to allow accumulation of debris below the level of the inlet and outlet pipes connecting them (the "side entry" nature of this system).  It prevents clogging of the outlet pipes and allows drainage.  This is why the sumps need to be cleaned out from time to time, as this paper shows with a figure.  So the sumps surely were part of the original design.  My question is: if this program eliminates the sumps and this important function, will the outlet pipes then not become clogged with debris over time, reducing their function and allowing stagnant water to accumulate, thus creating the kind of subterranean breeding that we wish to prevent?

Author Response

Dear Reviewer 

Many thanks for your feedback. Your first major concern is correct, there was no sampling study before and after the sumps were rectified to evaluate the program. The main purpose of the program was to determine if sumps were present in Darwin, and to remove them as potential larval habitat for exotic mosquitoes, hence the relatively limited sampling data and no investigation regarding their contribution to dry season mosquito problems in the suburbs. The intended conclusion is that side entry pit sumps no longer have to be considered a potential larval habitat in the event of an Ae. aegypti or Ae. albopictus incursion, with the side effect that some dry season endemic mosquito breeding was also remediated. I have updated the manuscript to hopefully allow this conclusion to be drawn. Overall, the mosquito engineering program is almost purely an operational program aimed at preventing breeding, hence the limited sampling data.

Your second major concern has been noted, and the manuscript updated to better conclude why there might have been earthern sumps in the three suburbs, but not elsewhere in Darwin at around the same time period (1950's-1980's) and during recent subdivisions in the past 10-15 years. Due to the high and intense rainfall conditions, and subsequent high velocity flows compared to other cities in Australia, it is the opinion of service authorities that side entry pit sumps would not be effective at trapping sediment, and thus there was the thought that scouring flows created the sumps rather than it being a design. The depths of the sumps were measured after desilting, and ranged from 5mm to 400mm deep, suggesting there was no standard design. A paragraph has been added to the manuscript discussion to outline the current methodology in Darwin with regards to managing sediment in stormwater drain systems and their end points, to ensure the stormwater pipe systems can be designed and maintained to be completely free draining. Blocked stormwater outlet pipes are likely to be of much greater mosquito productivity than the small sumps that were rectified, hence Medical Entomology and City of Darwin have the shared program to maintain pipe outlet drains, and other NT Government agencies also desilt their outlet drains when required. The manuscript has been updated to hopefully better explain why the sumps were rectified, and why there is a general absence of side entry pit sumps in Darwin.

I greatly appreciate your time taken in reviewing the manuscript.

Yours sincerely

Allan Warchot

Reviewer 2 Report

See attached file

Author Response

Dear Reviewer 

Many thanks for your feedback.

With regards to major comment 1, there was no sampling study before and after the sumps were rectified to evaluate the program, and whether the sumps were worthy of remediation. This was due to the operational based focus of the mosquito engineering program. It is most likely that residential habitat such as pot plant drip trays, bird baths, dog bowls and disused pools were of greater importance than the sumps, although Rapid Creek did contain a large number of sumps and perhaps there might have been a noticeable reduction in Cx. quinquefasciatus in that suburb. The main intention of the program was to remove the sumps as potential larval habitat for exotic container breeding mosquitoes, with the removal of dry season cryptic larval habitats for endemic mosquitoes another potentially positive outcome.  

With regards to major comment 2, yes the main focus is on the potential for the sumps to act as maintenance larval habitats for Ae. albopictus and Ae. aegypti, if an incursion was to occur. I have expanded the endemic mosquito discussion in the manuscript, although in the absence of before and after trapping data, the main take home message of the manuscript is still as mentioned above.

I have carried out the suggested edits provided in the minor comments, which can be seen as track changes in the revised manuscript. Many thanks for picking up on those errors.

I greatly appreciate your time taken in reviewing the manuscript.

Yours sincerely

Allan Warchot

Round 2

Reviewer 2 Report

no comments